# Silver Nanoparticle Targets Fabricated Using Chemical Vapor Deposition Method for Differentiation of Bacteria Based on Lipidomic Profiles in Laser Desorption/Ionization Mass Spectrometry

**DOI:** 10.3390/antibiotics12050874

**Published:** 2023-05-08

**Authors:** Ewelina Maślak, Adrian Arendowski, Michał Złoch, Justyna Walczak-Skierska, Aleksandra Radtke, Piotr Piszczek, Paweł Pomastowski

**Affiliations:** 1Centre for Modern Interdisciplinary Technologies, Nicolaus Copernicus University in Toruń, Wileńska 4 Str., 87-100 Toruń, Poland; e_maslak@doktorant.umk.pl (E.M.); aarendowski@umk.pl (A.A.); walczak-skierska@umk.pl (J.W.-S.); p.pomastowski@umk.pl (P.P.); 2Chair of Environmental Chemistry and Bioanalytics, Faculty of Chemistry, Nicolaus Copernicus University in Toruń, Gagarina 7 Str., 87-100 Toruń, Poland; 3Department of Inorganic and Coordination Chemistry, Faculty of Chemistry, Nicolaus Copernicus University in Toruń, Gagarina 7 Str., 87-100 Toruń, Poland; aradtke@umk.pl (A.R.); piszczek@umk.pl (P.P.); 4Nano-Implant Ltd., Gagarina 5/102, 87-100 Toruń, Poland

**Keywords:** bacteria, chemical vapor deposition, lipids, mass spectrometry, silver nanoparticles

## Abstract

The global threat of numerous infectious diseases creates a great need to develop new diagnostic methods to facilitate the appropriate prescription of antimicrobial therapy. More recently, the possibility of using bacterial lipidome analysis via laser desorption/ionization mass spectrometry (LDI-MS) as useful diagnostic tool for microbial identification and rapid drug susceptibility has received particular attention because lipids are present in large quantities and can be easily extracted similar to ribosomal proteins. Therefore, the main goal of the study was to evaluate the efficacy of two different LDI techniques—matrix-assisted (MALDI) and surface-assisted (SALDI) approaches—in the classification of the closely related *Escherichia coli* strains under cefotaxime addition. Bacterial lipids profiles obtained by using the MALDI technique with different matrices as well as silver nanoparticle (AgNP) targets fabricated using the chemical vapor deposition method (CVD) of different AgNP sizes were analyzed by the means of different multivariate statistical methods such as principal component analysis (PCA), partial least squares discriminant analysis (PLS-DA), sparse partial least squares discriminant analysis (sPLS-DA), and orthogonal projections to latent structures discriminant analysis (OPLS-DA). The analysis showed that the MALDI classification of strains was hampered by interference from matrix-derived ions. In contrast, the lipid profiles generated by the SALDI technique had lower background noise and more signals associated with the sample, allowing *E. coli* to be successfully classified into cefotaxime-resistant and cefotaxime-sensitive strains, regardless of the size of the AgNPs. AgNP substrates obtained using the CVD method were used for the first time for distinguishing closely related bacterial strains based on their lipidomic profiles and demonstrate high potential as a future diagnostic tool for the detection of antibiotic susceptibility.

## 1. Introduction

The development of new diagnostic methods that facilitate the initiation of targeted antimicrobial therapy through rapid and accurate identification of microorganisms is constantly driven by the global threat of numerous infectious diseases [1]. Among the methods that have significantly contributed to the transformation of the field of microbial diagnostics are mass spectrometry platforms targeting microbial products, mostly proteins [2,3]. In view of this, pathogens identification by matrix-assisted laser desorption/ionization time-of-fight mass spectrometry (MALDI-ToF MS) analysis of high abundance proteins (ribosomal among others) is emerging as the dominant technology in many clinical laboratories [4]. Although protein-based MALDI-ToF MS platforms revolutionized the workflow of routine clinical laboratories around the world, nevertheless, application of such approach meets some constraints such as a commonly encountered failure to differentiate some closely related species (e.g., *Escherichia coli* vs. *Shigella*), laborious protein extraction procedures required for microorganisms encased in complex, thick cell walls, poor efficiency in detecting pathogens directly in the clinical specimens as well as difficulty in differentiating antimicrobial-resistant strains [4,5,6]. Because, after proteins, lipids are the major functional and structural component of cells and play an important role in many cellular processes such as membrane formation, energy storage, and cell signaling, they are therefore characterized by a wide variety of characteristics, such as the backbone and the headgroups, the number of fatty acids, and the chemical moieties they are modified with [7,8]. The diversity of lipids, especially glycolipids which showed species-specific characteristics within bacteria, has the potential to make them useful biomarkers for microbial identification in much the same way that the protein-based platforms are used to identify bacteria [1,5]. Moreover, recent studies have shown that microbial lipid signatures generated during MS analysis can be even more powerful than those based on the proteomics [9,10,11]. Therefore, the possibility of using bacterial lipidome analysis as a complementary method to existing MALDI diagnostic platforms is currently receiving particular attention from researchers because bacterial lipids are present in large quantities and can be easily extracted similar to ribosomal proteins [4]. In addition, it has been noted that the lipid profiles of drug-resistant strains differ significantly from those of susceptible strains, regardless of whether the antibiotic action directly targets membrane lipids or other cellular components [12,13]. 

Despite the significant progress that has been made in lipid analysis using MS, there is still a great need for their further development, especially with respect to bioinformatics resources, including the creation of robust and accurate databases containing organism-specific lipid species to support clinical applications of bacterial lipidomics [14]. Moreover, it is known that the sensitivity of the MALDI-MS analysis for distinct classes of biomolecules can vary when employing different MALDI matrices. In view of this, Perry and colleagues [15] revealed significant matrix effects on the relative signal intensities observed for different lipid subclasses that brought about the generation of unique lipid profiles. It means that during the search for optimal conditions for the generation of lipid profiles, the matrix selection is a key factor that should be considered, especially as some lipid-derived compounds are exclusively detectable if the most appropriate matrix is used [16]. 

A technique that may be more appropriate for use in this type of analysis is surface-assisted laser desorption/ionization (SALDI), in which target plates contain various micro- or nanoscale structures [17,18]. Silver nanostructures are among the most commonly used nanomaterials in SALDI MS. They enable efficient absorption of laser radiation and energy transfer providing desorption of both ionic and non-ionic analyte molecules. Additionally, silver nanoparticles (AgNPs) exhibit a relatively high salt tolerance and the ability to accurately calibrate the internal spectrum on silver cluster signals [19]. Silver-assisted LDI has already been found to be a powerful cationization agent for MS analysis of various lipids and olefinic species such as cholesterol and fatty acids [20].

The main goal of the study was to differentiate cefotaxime-resistant and -sensitive *E. coli* strains based on their lipidomic profiles in laser desorption/ionization mass spectrometry using the MALDI technique and silver nanoparticle targets fabricated using the chemical vapor deposition method (CVD)—SALDI approach. *E. coli* is a model representative of Enterobacterales; the group demonstrated widespread resistance against beta-lactams, including cefotaxime (cephalosporins third generation belonging to last-resort medicaments), which is, therefore, crucial for the health care system and from the epidemiological point of view. It is known that distinguishing *E. coli* from its close relatives, including subspecies, differences in virulence level, and antibiotic resistance, can be problematic; however, as in the case of other Enterobacterales, resistance against cefotaxime can be detected using the MALDI approach based on the indication hydrolysis of the antibiotic molecules. Thus, the project aimed to check if such resistance can be detected using other LDI techniques, which analyze the bacterial lipidomic profiles. The performance of the classification was evaluated using different statistical methods—PCA, PLS-DA, sPLS-DA, and OPLS-DA. 

## 2. Materials and Methods

### 2.1. Bacterial Sample Preparation

Two strains of *Escherichia coli* isolated from diabetic foot were selected for analysis. The selected strains showed different sensitivity to β-lactam antibiotics, including cefotaxime. The resistance of isolates to this group of antibiotics was detected using the MALDI technique and confirmed using multiplex PCR. The cefotaxime-sensitive strain was designated as *E. coli* DFI30 [MZ960143], and the resistant strain as DFI4 [MZ918945] (manuscript under review). Strains were grown overnight in 5% sheep blood-supplemented medium (Columbia Blood Agar; Oxoid, UK) prior to analysis. A loop of 1 µL of bacterial biomass was suspended in 50 µL of antibiotic solution (0.5 mg/mL cefotaxime dissolved in MBT STAR Buffer). The samples were incubated with agitation (ca. 500 rpm) for 2 h at 37 °C. Incubation time was set following the manufacturer’s guideline for cefotaxime resistance detection using the MBT STAR BL assay to check whether this time is enough to distinguish isolates based on their lipidomic profiles. After incubation time, the samples were centrifuged (1 min, 13,000 rpm) and lipid extraction was performed from the biomass in the sediment. The extracts were prepared using the classical Folch method, a widely used standard method for lipid extraction in many laboratories. This method involves partitioning lipids in a biphasic mixture of chloroform and methanol. The bacterial biomass was suspended in 1.5 mL of a chloroform/methanol mixture (2:1, *v*:*v*) and placed in an ultrasonic bath for 10 min. An amount of 0.5 mL of 0.05 M NaCl was added to the sample and vortexed for 10 min. The samples were then centrifuged (15 min, 6000 rcf) and the chloroform layer was collected in a separate tube. Chloroform was evaporated and the obtained precipitate was dissolved in methanol and TA30 30% acetonitrile (CAN), 69.9% water, 0.1% trifluoroacetic acid (TFA). The samples prepared in this way were applied in an amount of 1 µL to the appropriate plates.

### 2.2. SALDI Targets Preparation

The stainless steel plates were covered with a silver coating consisting of densely packed silver nanoparticles (AgNPs) using a chemical vapor deposition (CVD) technique. Before the deposition process, the substrate surfaces were degreased by washing them in an ultrasonic bath with distilled water containing a non-ionic surfactant for 45 min (twice). Prepared plates were immersed in the acetone (analytical grade) for 30 min, the distilled water for 10 min, and after drying in an Ar stream, their surface was activated in 0.1% trifluoracetic acid solution for 20 min. After drying in an argon stream, the prepared substrate was placed in a CVD reactor. All CVD experiments were carried out using *hot wall* reactor (own construction) using the [Ag_5_(O_2_CC_2_F_5_)_5_(H_2_O)_3_] as a precursor. The synthesis and physico-chemical properties of the mentioned above precursor were earlier described [21,22]. Metallic AgNPs were deposited under conditions collected in Table 1.

The morphology of created surfaces with silver nanoparticles was studied using a scanning electron microscope (SEM, Quanta 3D FEG, Huston, TX, USA). The structure of AgNP coatings was investigated using an energy-dispersive X-ray diffractometer (Quantax 200 XFlash 4010, Bruker AXS GmbH, Karlsruhe, Germany) with a copper monochromator and CuKα radiation (λ = 0.15418 nm). XRD patterns were collected in the 2*Θ* range 10–80°, step 0.02°, and time 20 s. The Sartorius MCA2.7S-2S00-M microbalance (Sartorius Lab Instruments GmbH & Co. KG, Goettingen, Germany) was applied to determine the weight of the reference sample before and after the CVD process. The stainless steel (H17) reference samples of sizes 1 × 1 cm were placed in the CVD reactor together with the investigated sample to obtain similar deposition conditions.

For the purposes of the MALDI experiments, Ag films were prepared in real time, and the storage time of samples (closed box, room temperature, limited access to light) was not longer than 2–3 days.

### 2.3. Mass Spectrometry Analysis

MALDI measurements were performed on the Bruker 384 ground steel target. The matrices for MALDI analysis CHCA (α-Cyano-4-hydroxy-cinnaminic acid, Bruker Daltonik GmbH, Bremen, Germany), DHB (2,5-dihydroksybenzoic acid, Bruker Daltonik GmbH, Bremen, Germany), and super-DHB (90:10 2,5-DHB:2-hydroksy-5-methoxy-benzoic acid, Sigma Aldrich, Steinheim, Germany) were prepared according to the manufacturer’s recommended protocols (Bruker Daltonics GmbH, Bremen, Germany) as follows—(1) CHCA saturated in TA30, (2) 20 mg/mL DHB in TA30, and (3) 50 mg/mL super-DHB in TA50 (50:50 ACN:0.1% TFA in water). An amount of 0.5 µL of each sample was placed directly on the target and air-dried, then 0.5 µL of matrix was applied to the spots. We chose MALDI matrices offering sufficient sensitivities for lipids in positive ion mode MS analysis, among which DHB is recognized as the most common MALDI matrix in the lipid field [15].

SALDI targets after CVD were used with Bruker MTP Slide-Adapter II. Volumes of 0.5 µL of samples were placed directly on target plate, air-dried, and inserted into MS apparatus for measurements.

MS experiments were performed using a Bruker ultrafleXtreme time-of-flight mass spectrometer equipped with a SmartBeam II laser (355 nm and frequency 2 kHz) in positive ion reflectron mode. The measurement range was *m*/*z* 60–1500, and suppression was turned on for *m*/*z* lower than 59. The number of laser shots was 1500 (3 × 500 shots) for each sample spot. The first accelerating voltage was held at 25.08 kV and the second ion source voltage at 22.43 kV. Reflector voltages accounted for 26.64 and 13.54 kV. The value of detector gain for the reflector was 2.51×. The value of the global attenuator offset accounted for 30%. Mass calibration was performed with FlexAnalysis 3.3 (Bruker Daltonics GmbH) using the cubic enhanced model and internal standards, silver ions and clusters from Ag+ to Ag_4_+ for SALDI targets and matrix peaks for MALDI. The mass list for each sample was created using the ‘centroid’ peak detection algorithm and signal to noise (S/N) threshold equal to 3.

### 2.4. Analysis of MS Results

Statistical analysis of the results was performed with the use of the MetaboAnalyst 5.0 web service [23]. The interquantile range method was used for data filtering. Data were normalized by sum, cube root transformed, and default Pareto scaling was applied. Principal component analysis (PCA), partial least squares discriminant analysis (PLS-DA), sparse partial least squares discriminant analysis (sPLS-DA), orthogonal projections to latent structures discriminant analysis (OPLS-DA), and analysis of variance (ANOVA) statistical methods were used. In addition, a random forest classification method and hierarchical clustering dendrograms with Euclidean distance measures and the Ward clustering algorithm were applied.

## 3. Results

The use of the selected conditions during the CVD process resulted in the production of the coatings consisting of the dense-packed silver particles uniformly covering the entire surface of the stainless steel plates (Figure 1A). The use of different masses of solid Ag precursor under similar deposition conditions (Table 1) allowed the control of the surface morphology of the deposited layers as well as the size of the deposited AgNPs. Layers, which consisted of dense-packed silver nanoparticles, similar in shape to a sphere, were produced in the case of low precursor concentrations in vapors (precursor weight—5–15 mg). Medium grain sizes of AgNPs changed from 50 ± 10 up to 240 ± 80 nm for coatings produced using 5 and 15 mg of the precursor, respectively (Table 2). The metallic silver was identified basing the reflection peaks, which were found at 38.2°, 44.3°, 64.6°, and 77.6° assigned to planes indexed as (111), (200), (220), and (311), respectively (Figure 1B) [24]. 

The precursor weight and the deposition time (1h) guarantee the deposition of uniform monolayers (over the entire surface of the substrate), composed of the spherical AgNPs. Analysis of data presented in Table 2 showed direct dependency between the weight of the used precursor and the size of deposited grains. Maintaining the repeatability of both factors mentioned above allows for precise control of the size of the deposited AgNPs and their packing density.

Figure 2 shows the results of statistical analysis of data from measurements of lipid extracts of strains DFI4 and DFI30 by the MALDI-MS technique using three matrices: α-cyano-4-hydroxycinnamic acid (CHCA), 2,5-dihydroxybenzoic acid (DHB), and super-DHB. From analyzing charts, it can be concluded that only the sPLS-DA score plot (Figure 2C) presents completely separated groups. The other three used statistical methods, PCA (Figure 2A), PLS-DA (Figure 2B), and OPLS-DA (Figure 2D), did not allow for complete separation of analyzed strains. In Figure 2B,D, it can be seen that the samples that most affect the separation of groups are the extracts measured using the CHCA matrix. Therefore, it can be concluded that the matrix used has a decisive effect on the separation of groups in the MALDI method. This is confirmed by the dendrogram made for the same data (Figure 3B). 

The large effect of the method can also be seen in the statistical analysis of all results divided into two groups, the DFI4 and DFI30 strains (Figure 4), and four groups, DFI4-SALDI, DFI4-MALDI, DFI30-SALDI, and DFI30-MALDI (Figure 5). None of the statistical methods allowed the complete separation of strains when two groups were used. The data that determined this were samples from MALDI measurements (Figure 4). When four groups were used, extracts from strains DFI4 and DFI30 measured by MALDI were clustered completely (Figure 5A), or almost completely overlapping (Figure 5B,C). Analysis of the VIP scores (variable importance in projection) showed remarkable differences between the LDI method used regarding significant *m*/*z* signals that distinguished examined isolates (Appendix A). Furthermore, the ANOVA test showed that, considering the nine most discriminating *m*/*z* values obtained for DFI4 and DFI30 bacterial lipids samples, in the case of the SALDI approach, the variation between *E. coli* isolates was much higher than in the case of the MALDI one (Appendix A). These results suggest a high degree of method influence on the spectral data over the sample data for the MALDI-MS technique. Such a phenomenon was reflected in the set of lipids that differentiate investigated *E. coli* isolates which differ considerably between MALDI and SALDI (Table 3). 

The SALDI method based on plates with silver nanostructures applied by the CVD technique shows a complete separation of the two tested strains by all the statistical methods used (Figure 6). In addition, the generated dendrogram first shows the grouping of samples by strain and then by the data recorded on the different plates (Figure 3A). Additionally, when all results are analyzed across two or four groups, data from SALDI-MS measures show more variation than data from MALDI (Figure 4 and Figure 5). This implies a smaller effect of the SALDI plates used on the spectral data than is the case with the MALDI method. This can be explained by the presence of less chemical background in the form of additional signals from the matrix used (Table 4). 

In the case of SALDI wafers coated with silver nanostructures using the CVD technique, only three groups of high-intensity signals originating from Ag^+^ to Ag_3_^+^ silver clusters are present on the spectra, while in the MALDI method there is a significant number of peaks originating from the applied matrix (Appendix A).

## 4. Discussion

The clustering of *E. coli* lipid extracts observed during the study mainly depends on the matrix used (MALDI method) and may be due to interference of peaks from the matrix. It is known that standard MALDI matrices are low-molecular-weight organic acids that produce a variety of matrix-related ions, which cause interfering signals in the low-mass range as a consequence [25]. Indeed, analysis of the blank MS spectra performed for SALDI targets and MALDI matrices showed that the latter contain significantly more signals than the former. Moreover, extracts analyzed using the CHCA matrix, characterized by a significantly higher number of signals from the matrix itself (307, S/N ≥ 3), showed a more distinctive MS pattern than the other two matrices. This effect may also explain the high similarity between the MS profiles of the tested *E. coli* strains when using the CHCA matrix. Such phenomenon was not observed in the case of SALDI targets with different AgNP sizes, where differences in the number of signals derived from blank MS spectra were comparatively minor. On the contrary, both investigated bacterial strains were clearly divided regardless of the silver particles’ size. The observed phenomenon may be due to both lower background noise and more MS signals generated from the sample itself (ca. two times more compared to the MALDI mode). Similar results were obtained for metabolomic profiling of two mold species using the SALDI target, with gold nanoparticles obtaining more than three times as many signals and matched metabolites compared to the MALDI method [26]. As Hansen and colleagues noted [27], the nanosubstrates employed in SALDI-MS experiments offer a clear background in the low *m*/*z* range, thus favoring the analysis of small molecules such as lipids (mostly < 1000 Da). In addition, the observed higher number of peaks in the SALDI variant may arise from its higher efficiency in the detection of lipids with low ionization efficiency, such as neutral lipids. Indeed, silver nanoparticles were proven to be effective substrates for analyzing a wide range of the lipids such as glycerolipids, sterol lipids, fatty acids, and sphingolipids in positive ionization mode [28]. Furthermore, it was shown that the detection of neutral lipids, which can be challenging in conventional MALDI-MS measurements due to ion suppression caused by phospholipids, can be easily performed using the SALDI method [29,30]. The mass spectrometry imaging of *Bacillus subtilis* colony biofilms by Lukowski and colleagues [31] revealed that different ionization techniques (metal-assisted vs. MALDI) led to the identification of a unique subset of molecular species, where the application of the Au-covered targets enabled the identification of more small molecules and neutral lipids compared to the use of the DHB matrix. 

Chemical vapor deposition (CVD) belongs to the targets fabrication strategy that allows exact control of the nucleation and growth of metallic layers with strictly defined structure, morphology, physicochemical and biological properties, and is also characterized by high purity [22]. Indeed, all SALDI targets obtained were characterized by uniform coatings with silver particles of similar shapes and did not exhibit the sweet-spot phenomenon. Such characteristics may explain the similar classification of lipid extracts depending on the bacterial strain, despite the observed differences in the size of AgNPs between the prepared SALDI targets. It may also suggest that the morphology of the silver substrate had a more significant effect on SALDI-MS performance than the size of the AgNPs concerning lipid profiling, which is consistent with the common claim that the nature and morphology of the substrate have a significant effect on the ionization efficiency of the analytes [32]. Nevertheless, the size-dependent phenomenon of the ionization efficiency of small-molecule compounds during SALDI-MS analysis has been previously reported in the literature and found to be significant [33]. As for silver, Ding et al. [34] found that AgNPs of various sizes had different ionization efficiencies for amyloid-beta peptides. The authors noted that the small-size AgNPs (2.8 ± 1.0 nm) showed the best SALDI-MS performance, where ion yields were nearly two- and fourfold higher than those from the 12.8 ± 3.2 nm and the 44.2 ± 5.0 nm AgNPs, respectively. The observed phenomenon was considered to be related to differences in the specific surface area of AgNPs—the larger the surface area, the more peptide molecules, and laser energy could be absorbed.

Our study showed that a 2 h incubation with the addition of an antibiotic (cefotaxime) was sufficient to separate resistant and susceptible *E. coli* strains based on their lipidomic profiles. Similarly, Liang et al. [1] investigated that Gram-negative bacteria belonging to the ESKAPE pathogens (*E. coli*, *K. pneumoniae*, *P. aeruginosa*, and *P. mirabilis*) could distinguish between colistin-susceptible and -resistant strains using MALDI-MS analysis based on the presence of colistin-resistance-associated ions. The same authors in another paper [4] showed that antimicrobial-resistant ESKAPE isolates are chemically distinct from their susceptible counterparts. This led to clustering together colistin-resistant *K. pneumoniae* and *A. baumannii* strains based on the mass spectra of their glycolipids during a performed hierarchical cluster analysis. It should be mentioned that both works are related to the detection of the antimicrobial resistance that is mediated via direct modification of membrane lipids—lipid A in this case—thus the observed differences in the profiles of the lipids studied were limited to mass shifts in the lipid A molecules. Nevertheless, Schenk and colleagues [35] found a correlation of lipid profiles with sensitivity to antibiotics other than polymyxins. The authors revealed a variation in lipid content (altered distribution of the fatty acids and glycerophospholipids) in the closely related strains of *E. coli* exposed to sublethal concentrations of norfloxacin using MALDI coupled to Fourier transform ion cyclotron resonance mass spectrometry (FT-ICR). In turn, Xie et al. [36] used multiple reaction monitoring profiling (MRM profiling) of lipids to distinguish strain-level differences in microbial resistance in *E. coli* that had been cultured with and without beta-lactam antibiotic exposure. The MRM-profiling method was able to distinguish between resistant and non-resistant *E. coli* strains treated with amoxicillin or amoxicillin with clavulanate at 1/2× and 1× the minimum inhibitory concentration. Our study showed that cefotaxime-resistant and -sensitive *E. coli* strains also show a different lipid pattern when cultured in the presence of this antimicrobial agent. However, using a matrix, especially CHCA, hinders the grouping of the strains. On the contrary, using the SALDI technique with silver particle coating overcame the obstacles encountered during MALDI analysis regardless of the size of the AgNPs, which was possible by obtaining homogeneous silver films using the CVD technique. To our knowledge, this is the first work related to applying the SALDI technique for distinguishing sensitive and resistant bacterial strains based on their lipidomic profiles. The SALDI method outperformed the MALDI technique in small molecule analysis for closely related strains’ discrimination. It should be further investigated to find a suitable diagnostic tool for early drug resistance detection.

## 5. Conclusions

Lipidomic profiles of cefotaxime-resistant and cefotaxime-sensitive *E. coli* strains obtained by laser desorption/ionization mass spectrometry analysis differ as influenced by the presence of the antibiotic in the culture medium. MALDI mode clustering of lipid profiles is hampered by interference with matrix-derived ions, particularly CHCA ions. This obstacle was overcome by using the SALDI technique, in which CVD-fabricated silver nanoparticle targets allowed the strains to be clearly grouped regardless of the size of the AgNPs. SALDI AgNP targets obtained via the CVD method are characterized by low background noise and more MS signals derived from the samples, most probably related to the higher lipid ionization efficiency. AgNP substrates obtained using the CVD method were used for the first time for distinguishing closely related bacterial strains based on their lipidomic profiles and demonstrate high potential as a future diagnostic tool for detection of antibiotic susceptibility.

## Figures and Tables

**Figure 1 antibiotics-12-00874-f001:**
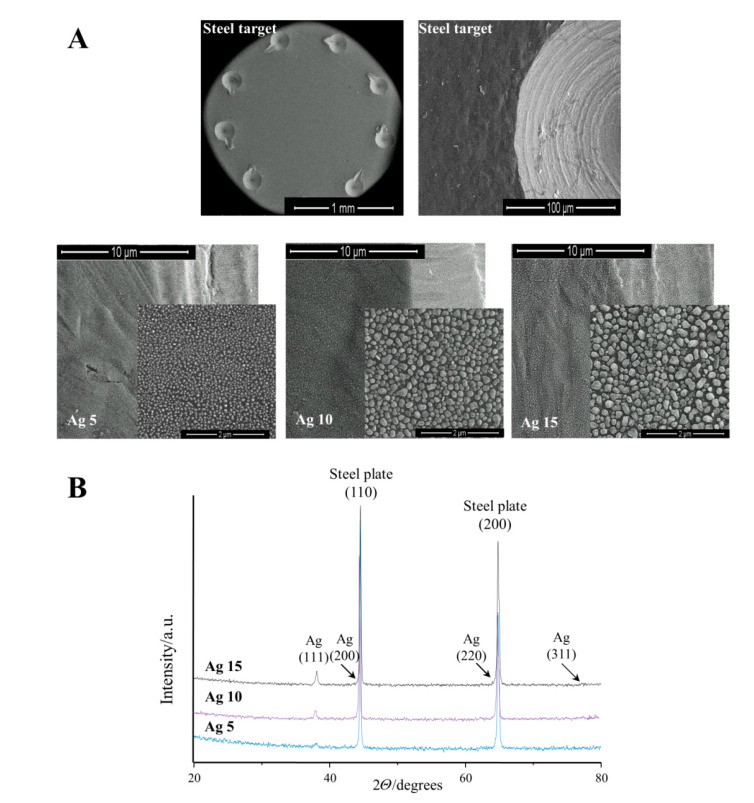
SEM images (**A**) of the bare plates as well as silver coatings obtaining using the CVD process with different amounts of the precursor (5, 10, and 15 mg). Registered XRD patterns for the studied plates (**B**) confirmed the deposition of the silver nanoparticles on the surface of the steel substrates. Ag 5, 10, 15—silver coating obtained using 5, 10, and 15 mg of the precursor; a.u.—arbitrary units.

**Figure 2 antibiotics-12-00874-f002:**
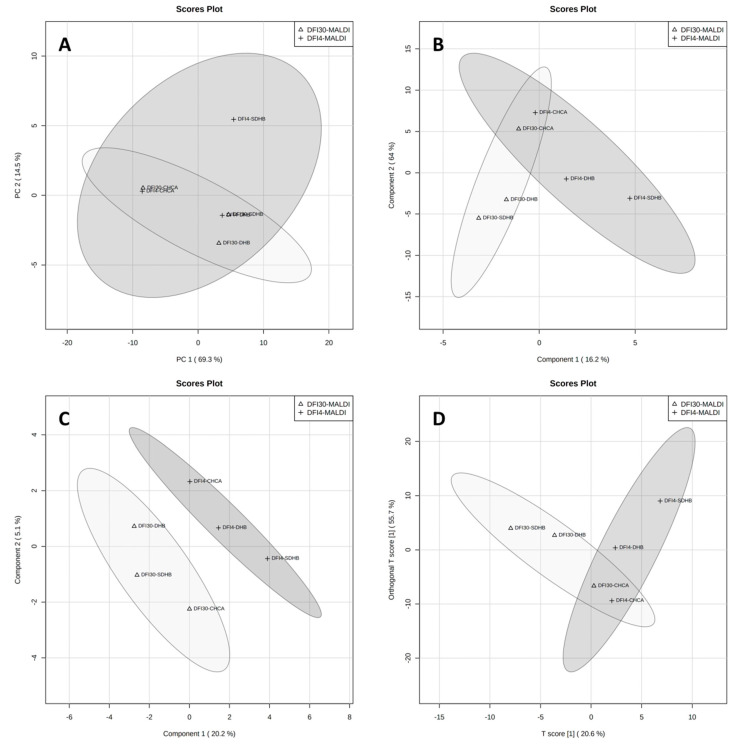
Graphical representation of statistical analysis of MS data from MALDI experiment: PCA component 1 vs. 2 (**A**), PLS-DA component 1 vs. 2 (**B**), sPLS-DA component 1 vs. 2 (**C**), and OPLS-DA (**D**). Dark-grey area represents data for DFI-4 strain while light-grey represents DFI-30 strain.

**Figure 3 antibiotics-12-00874-f003:**
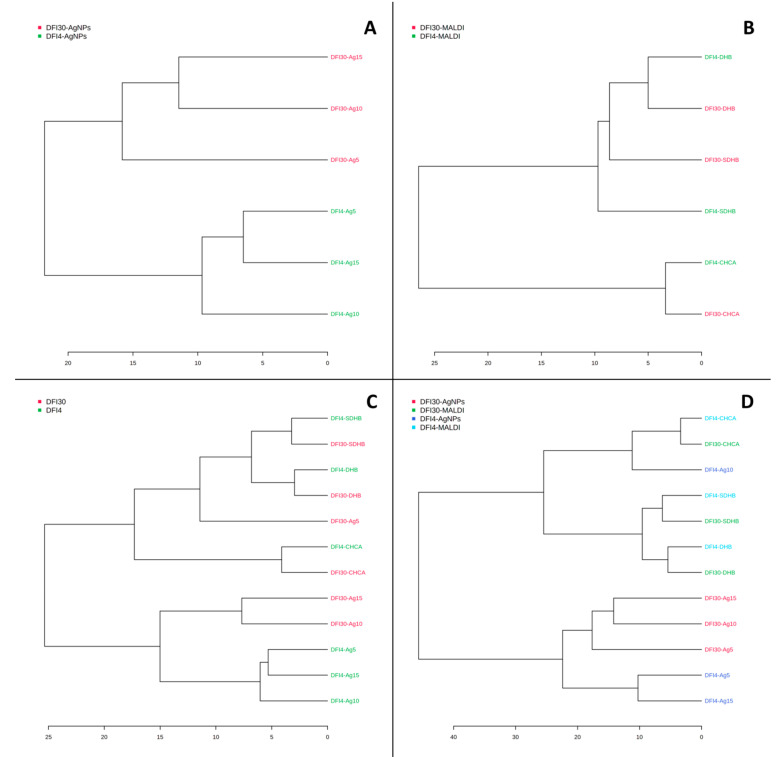
Dendrograms for MS data from SALDI/NALDI (**A**), MALDI (**B**) experiments and with the use of two groups (**C**) and four groups (**D**) for all data.

**Figure 4 antibiotics-12-00874-f004:**
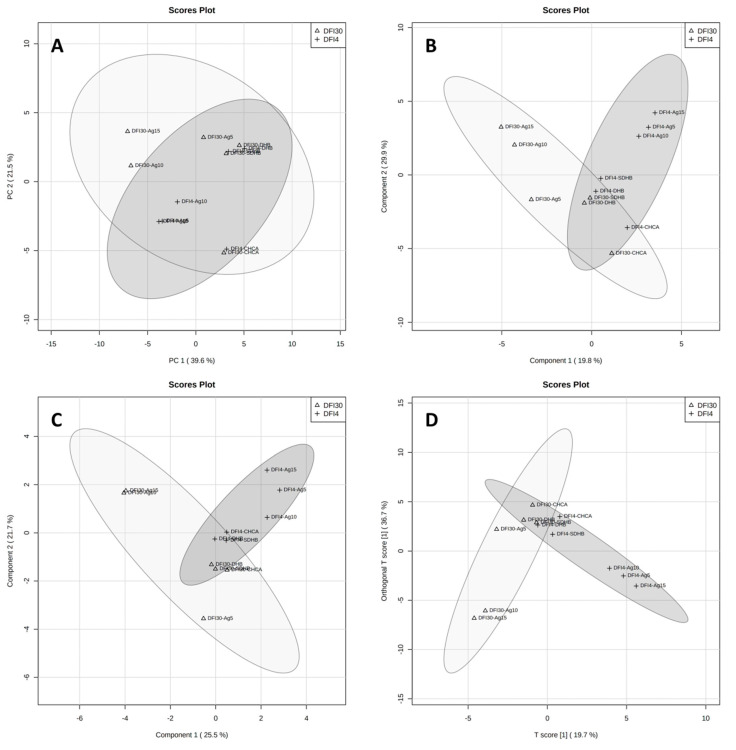
Graphical representation of statistical analysis of MS data from MALDI and SALDI/NALDI experiment with two groups: PCA component 1 vs. 2 (**A**), PLS-DA component 1 vs. 2 (**B**), sPLS-DA component 1 vs. 2 (**C**), and OPLS-DA (**D**). Dark-grey area represents data for DFI-4 strain while light-grey represents DFI-30 strain.

**Figure 5 antibiotics-12-00874-f005:**
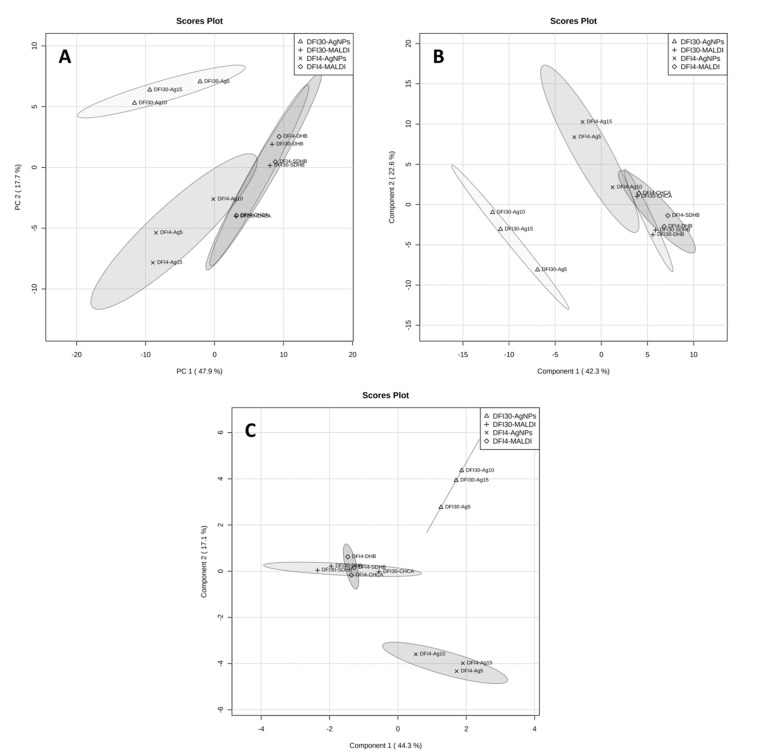
Graphical representation of statistical analysis of MS data from MALDI and SALDI/NALDI experiment with four groups: PCA component 1 vs. 2 (**A**), PLS-DA component 1 vs. 2 (**B**), and sPLS-DA component 1 vs. 2 (**C**).

**Figure 6 antibiotics-12-00874-f006:**
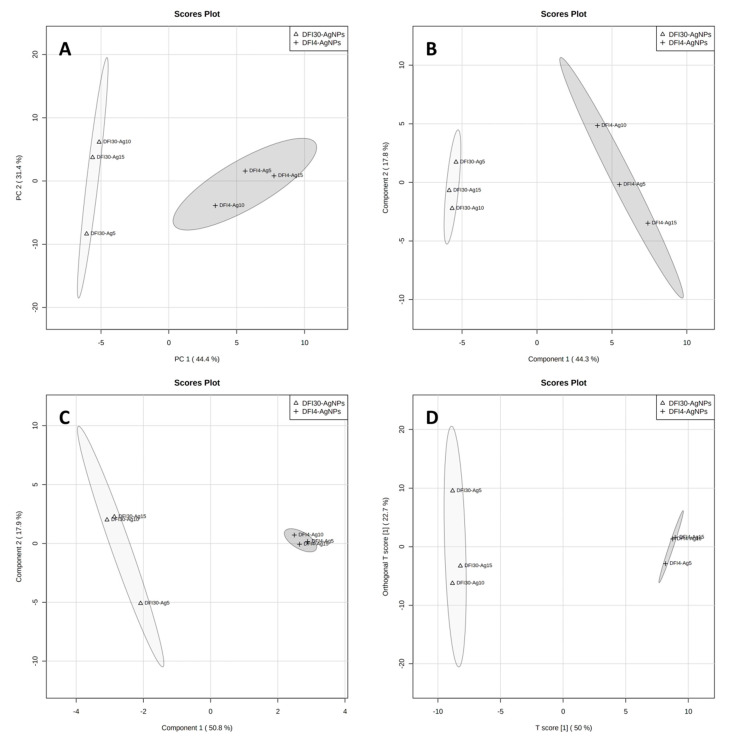
Graphical representation of statistical analysis of MS data from SALDI/NALDI experiment: PCA component 1 vs. 2 (**A**), PLS-DA component 1 vs. 2 (**B**), sPLS-DA component 1 vs. 2 (**C**), and OPLS-DA (**D**). Dark-grey area represents data for DFI-4 strain while light-grey represents DFI-30 strain.

**Table 1 antibiotics-12-00874-t001:** Deposition parameters AgNP coatings.

Precursor	Ag_5_(O_2_CC_2_F_5_)_5_(H_2_O)_3_
Precursor weight (mg)	5, 10, 15
Vaporization temperature (T_V_) (°C)	230
Carrier gas	Ar
Total reactor pressure (p) (mbar)	3, 0
Substrate temperature (T_D_) (°C)	290
Substrates	stainless steel (H17)
Deposition time (min)	60
Sample heating time (min)	30 (Ar/H_2_ (3:1%)

**Table 2 antibiotics-12-00874-t002:** AgNP films deposited by CVD technique.

Target	Precursor Weight (mg)	Percentage Substrate Mass Increase after the CVD Process (wt.%)	AgNPs Medium Grain Size (nm)
Ag 15	15	0.06	240 ± 80
Ag 10	10	0.04	150 ± 50
Ag 5	5	0.03	50 ± 10

**Table 3 antibiotics-12-00874-t003:** List of ions and lipids found by PLS-DA statistical analysis of bacterial lipids samples’ mass spectra.

Lipid	Ion Formula	Experimental *m*/*z*	Calculated *m*/*z*	Reg. ^a^	
				DFI4	DFI30
SALDI					
FA 6:0;O	[C_6_H_12_O_3_ + H]^+^	133.074	133.0859	+	-
FA 5:1	[C_5_H_8_O_2_ + Na]^+^	123.132	123.0416	+	-
FA 6:3;O3	[C_6_H_6_O_5_ + H]^+^	158.963	159.0288	+	-
FA 30:1;O2	[C_30_H_58_O_4_ + K]^+^	521.409	521.3967	+	-
LPG 17:1	[C_23_H_45_O_9_P + K]^+^	535.1725	535.2433	+	-
FA 7:1	[C_7_H_12_O_2_ + Na]^+^	151.075	151.0729	+	-
FA 4:1	[C_4_H_6_O_2_ + K]^+^	124.936	124.9999	+	-
MALDI					
CAR 3:1	[C_10_H_17_NO_4_ + H]^+^	216.052	216.1230	+	-
FA 6:2;O4	[C_6_H_8_O_6_ + K]^+^	214.9205	214.9952	+	-
TG 64:3	[C_67_H_124_O_6_ + H]^+^	1026.025	1025.9471	-	+
CoA 4:1;O2	[C_25_H_40_N_7_O_19_P_3S_ + H]^+^	868.085	868.1385	+	-
PC 16:4	[C_24_H_40_NO_8_P + K]^+^	540.132	540.2123	+	-
FA 6:2;O3	[C_6_H_8_O_5_ + K]^+^	198.955	199.0003	+	-
CAR 14:1;O2	[C_21_H_39_NO_6_ + K]^+^	440.173	440.2409	+	-

^a^ Regulation of the intensity in samples. FA—fatty acid; LPG—lyso-phosphatidylglycerol; CAR—acyl carnitine; TG—triacylglycerol; CoA—acyl CoA; PC—glycerophosphocholine.

**Table 4 antibiotics-12-00874-t004:** Number of the signals in MS spectra depending on the LDI method used—MALDI vs. SALDI.

	Signals Number (S/N ≥ 3)		
Matrix/Nanostructures	Blank MS	*E. coli* DFI4	*E. coli* DFI30
DHB	90	262	241
Super-DHB	88	177	144
CHCA	307	447	485
Ag5	65	462	402
Ag10	47	366	315
Ag15	56	399	358

## Data Availability

The data presented in this study are available on request from the corresponding author. The data are not publicly available due to the need to access the commercial software used to generate data.

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
