# Peer review of "Silver Nanoparticle Targets Fabricated Using Chemical Vapor Deposition Method for Differentiation of Bacteria Based on Lipidomic Profiles in Laser Desorption/Ionization Mass Spectrometry"

_antibiotics, 2023, doi:10.3390/antibiotics12050874_

Round 1

Reviewer 1 Report

Comments for Authors

The authors performed mass spectrometry analysis of lipidome of two E. coli strains using two different LDI-MS methods, MALDI and SALDI approach. They succeeded in showing that SALDI approach is more efficient in distinguishing differences between strains. This result will contribute to the development of diagnostic techniques for strains, including AMR strains.

The experiment was done properly and the results were good, but I think that the value of this experiment would be even greater if the data analysis and its interpretation were improved.

The issues that the authors should address (if possible) are listed below.

1) Verify that the constructed random forest model can distinguish other strains as cefotaxime sensitive or resistant. If it does not done, I think there is no point in applying random forest. This is because it is natural that random forests can predict to two groups that have already been separated by PCA, which is a linear model.

2) Interpretation of the results of the multivariate analysis. In particular, the authors should confirm what kind of compounds have a high factor load of the first principal component in Fig. 6A and which compounds have a high VIP score in PLS_DA (Fig. 6B). It should then be considered whether compounds contributing to group differentiation are likely to be associated with antibiotic resistance. Without this analysis, it is not possible to distinguish from the results whether DFI4/DFI30 are clustered by the antibiotic resistance or simply by strain differences. I have no experience with mass spectrometry, so I'm sorry if I've made an impossible suggestion.

3) Fig. 5D is completely uninterpreted. Here, I think this analysis is meaningless unless authors look at what kind of compounds have significance, i.e. compounds where the abundance of these compounds is significantly different between groups.

In this study, authors have done a good job of demonstrating that SALDI approach can distinguish strains that could not be distinguished by MALDI. I don't think there is any evidence that strains can be classified "depending on their susceptibility to cefotaxime", described in the line 23.

Here are minor points:

Line 178: "used" might be "use".

Line 188: the peak at 77.6° is not assigned as (311) in Figure 1B.

Table 2: "AgPs" might be "AgNPs".

Corresponding part of Fig.S1 is not shown in the text

Author Response

Comments for Authors

The authors performed mass spectrometry analysis of lipidome of two E. coli strains using two different LDI-MS methods, MALDI and SALDI approach. They succeeded in showing that SALDI approach is more efficient in distinguishing differences between strains. This result will contribute to the development of diagnostic techniques for strains, including AMR strains.

The experiment was done properly and the results were good, but I think that the value of this experiment would be even greater if the data analysis and its interpretation were improved.

The issues that the authors should address (if possible) are listed below.

Question 1: Verify that the constructed random forest model can distinguish other strains as cefotaxime sensitive or resistant. If it does not done, I think there is no point in applying random forest. This is because it is natural that random forests can predict to two groups that have already been separated by PCA, which is a linear model.

Answer: Thanks Reviewer for the comment. We agree with the suggestion that there is no point in applying random forest since another method, like PCA, gave us to answer about grouping the investigated isolates. Thus, the sentence regarding the random forest model has been deleted to avoid results description repetition.

Question 2: Interpretation of the results of the multivariate analysis. In particular, the authors should confirm what kind of compounds have a high factor load of the first principal component in Fig. 6A and which compounds have a high VIP score in PLS_DA (Fig. 6B). It should then be considered whether compounds contributing to group differentiation are likely to be associated with antibiotic resistance. Without this analysis, it is not possible to distinguish from the results whether DFI4/DFI30 are clustered by the antibiotic resistance or simply by strain differences. I have no experience with mass spectrometry, so I'm sorry if I've made an impossible suggestion.

Answer: Thanks Reviewer for a suggestion. VIP scores in the PLS-DA method are shown in Figure S1. A search of the LIPID MAPS database was also performed and the result of the putative identification of the lipids is presented in Table 4. E. coli is considered a problematic bacterial species that are hard to differentiate based on its proteome and genome, not only regarding different subspecies or resistant/sensitive isolates but also from other closely related species like Shigella sp.. However, it is known that MBT STAR BL assay enabled classification of the E. coli strains on beta-lactams resistant/sensitive based on the antibiotic hydrolysis products detection. Therefore, we would like to check if such diversification could be also done but using analysis of the bacterial lipidomic profiles. In view of this, the main goal of this study was to compare different LDI conditions and approaches to find the best analytical solution. Based on the obtained results, we can suppose that observed diversification under antibiotic pressure can be potentially associated with the level of antibiotic resistance; however, we agree with the Reviewer's comment that further deepening analysis, including precise lipids identification –also in control conditions – should be performed to clarify if the observed phenomenon is mainly related to drug resistance level or to strain-dependent differences.

Question 3: Fig. 5D is completely uninterpreted. Here, I think this analysis is meaningless unless authors look at what kind of compounds have significance, i.e. compounds where the abundance of these compounds is significantly different between groups.

Answer: Thanks Reviewer for your valuable comment. We agree with the reviewer. Figure 5D has been replaced by Figure S2 with box plots for the nine most discriminating m/z values obtained by ANOVA analysis.

Question 4: In this study, authors have done a good job of demonstrating that SALDI approach can distinguish strains that could not be distinguished by MALDI. I don't think there is any evidence that strains can be classified "depending on their susceptibility to cefotaxime", described in the line 23.

Answer: Thanks Reviewer for your insightful comment. The sentence has been changed to avoid overestimation of the results. Although there is a rationale behind the conclusion that the observed differences are related to different sensitivity to cefotaxime, they require confirmation in further studies focused not on the search for optimal conditions to capture differences between isolates but on the contribution of lipids to the response to the presence of the antibiotic, which is planned for the future.

Here are minor points:

Question 4: Line 178: "used" might be "use".

Answer: Done.

Question 5: Line 188: the peak at 77.6° is not assigned as (311) in Figure 1B.

Answer: The mistake has been corrected. Peak at 77.6° was present in the Figure, however, its intensity is low due to limited sensitivity of the apparatus used, thus, we forgot to marked it in the previous Figure 1B.

Question 6: Table 2: "AgPs" might be "AgNPs".

Answer: Done.

Question 7: Corresponding part of Fig.S1 is not shown in the text

Answer: Corresponding part of Fig.S1 (in revised version changed to Fig.S3) is actually present in the text – please see page 7 – “In the case of SALDI wafers coated with silver nanostructures using the CVD technique, only three groups of high-intensity signals originating from Ag+ to Ag3+ silver clusters are present on the spectra, while in the MALDI method there is a significant number of peaks originating from the applied matrix (Fig. S3).”

Reviewer 2 Report

English is fine. Minor spell check required.

Author Response

In this article et al. discussed about the evaluation of the efficacy of two different  LDI  techniques  -  matrix-assisted  (MALDI)  and surface-assisted  (SALDI) approaches, in the classification of the closely related  Escherichia coli  strains depending on their susceptibility to cefotaxime. Various studies were performed. The specific comments about the article are given below.

Comments to Authors

Question 1.: Please carefully check the manuscript for spelling and contextual errors.

Answer: Done.

Question 2: Why authors choose E.coli species and why only cefotaxime antibiotic?   it better to prove the accuracy of the two methods using two or more different species or antibiotics? And there is no mention about the control lipidomic profile (without the drug)? What were the differences among all these? The lipid composition should be listed in a table and also mentioned any differences?  Does the signal generation has any relation to the concentration or type of lipids?

Answer: Thanks to the Reviewer for the valuable comments. E. coli is a model representative of Enterobacterales - the group of bacteria that demonstrated widespread resistance against beta-lactams, including cefotaxime which belongs to last-resort medicaments and therefore is crucial for the health care system. Since resistance against cefotaxime can be detected within Enterobacterales using the MALDI approach based on the indication hydrolysis of the antibiotic molecules, thus, the project aimed to check if the presence of such resistance can also be detected using other LDI techniques, which is an analysis of the bacterial lipidomic profiles. Cefotaxime was chosen for this study because investigated isolates varied in resistance level against this drug, which was checked in another study using MBT STAR BL assay. Secondly, cefotaxime belongs to the cephalosporins 3rd generation – crucial antibiotics used to eradicate gram-negative bacteria characterized by drug resistance against commonly used beta-lactams. Therefore, detection resistance against cefotaxime is crucially essential information for clinicians given proper drug treatment administration and preventing widespread resistance. Suitable explanation has been added to the manuscript – “E. coli is a model representative of Enterobacterales - the group demonstrated widespread resistance against beta-lactams, including cefotaxime (cephalosporins 3rd generation belonging to last-resort medicaments), which is, therefore, crucial for the health care system and from the epidemiological point of view. It is known that distinguishing  E. coli from its close relatives, including subspecies, differences in virulence level, and antibiotic resistance, can be problematic; however, as in the case of other Enterobacterales, resistance against cefotaxime can be detected using the MALDI approach based on the indication hydrolysis of the antibiotic molecules. Thus, the project aimed to check if such resistance can be detected using other LDI techniques, which analyze the bacterial lipidomic profiles.”

Since the project aimed to check if the differences in the bacterial lipidomic profiles under drug presence can be used much in the same way like detection of beta-lactam resistance using MALDI techniques and standard MBT STAR BL assay, we used the same conditions as in the mentioned MBT STAR BL assay in which bacterial suspension is incubated with the antibiotic addition and no control conditions are applied. Moreover, a comparison of the distinct LDI techniques has been performed to evaluate the efficiency of both approaches. We agree with Reviewer that analysis of the lipidomic profiles of the isolates under control conditions will be essential to check what kind of changes in the lipids compositions underwent when cefotaxime was added. Such information will be crucial to draw any conclusions about possible resistance mechanisms associated with bacterial lipids, which will be the subject of further studies focused on explaining the possible resistance mechanisms. Nevertheless, since performed analysis allowed us to check which lipids were primarily responsible for the division of the investigated E. coli strains on the resistant and sensitive to cefotaxime (based on the VIP scores), we added information about identifying them and their role in the bacterial cells.

As we mentioned in the manuscript, the type of the LDI technique used (silver nanoparticles, type of MALDI matrices) is supposed to have a pivotal impact on the generated lipid signals and their intensities. This is one of the most crucial founding of our work - to check which conditions will be more favorable for distinguishing investigated isolates. Further studies will focus on the more profound analysis of the individual lipids that demonstrated the most distinguishing power. Such studies will include also control conditions – without antibiotic addition – to check if differential lipids can be associated with the drug resistance mechanism. 

Question 3: How was the drug incubation time determined? Why only 2 hours?

Answer: Thanks to the Reviewer for the insightful remarks. As the MALDI technique can be used to assess E. coli resistance to cefotaxime based on the detection of the antibiotic's hydrolysis products, where the incubation time of the bacterial suspension with the antibiotic solution is two hours, hence the same time was chosen for the experiment in this study to see if it was sufficient to show differences between resistant and sensitive isolates but based on differences in lipidome. To clarify this, suitable description has been added to the M&M section.

Question 4: Does the thickness of the silver nanoparticles layers will have any effect?  And  how to exactly  control  the  thickness  of  the  layers  without  affecting  its  size  to  increase  its reproducibility?

Answer: Thanks Reviewer for the valuable comment. The precursor weight and the deposition time (1h) guarantee the deposition of uniform monolayers (over the entire surface of the substrate), composed of the spherical AgNPs. Analysis of data presented in Table 2 showed direct dependency between the weight of the used precursor and the size of deposited grains. Maintaining the repeatability of both factors mentioned above allows for precise control of the size of the deposited AgNPs and their packing density. Such explanation has been added to the manuscript.

Reviewer 3 Report

Line 120:

Authors should have described the CVD process in greater detail. For example the apparatus used.

Line 143 to 146:

It would have been beneficial for the authors to include norharmane matrix in addition to the commonly used CHCA, DHB, and super-DHB matrices. Although these matrices are widely used, it is unclear if they are the most suitable for analyzing bacterial membrane lipids. Norharmane matrix has been previously utilized for colistin resistance, as reported in the literature. The authors are advised to provide a rationale for using CHCA, DHB, and super-DHB matrices in the discussion section.

References on usage of norhormane:

1.       https://doi.org/10.1128/spectrum.01445-21

2.       https://doi.org/10.1038/s41598-020-78401-3

3.       https://doi.org/10.1093/femspd/ftw097

Line 178:

Typo: The use instead of The used.

Author Response

Question 1: Line 120: Authors should have described the CVD process in greater detail. For example the apparatus used.

Answer: Thank’s the Reviewer for the valuable comment. The description regarding CVD process has been rewritten [see section SALDI targets preparation] and providing all details related to the type of the used apparatus as well as procedures steps.

Question 2: Line 143 to 146: It would have been beneficial for the authors to include norharmane matrix in addition to the commonly used CHCA, DHB, and super-DHB matrices. Although these matrices are widely used, it is unclear if they are the most suitable for analyzing bacterial membrane lipids. Norharmane matrix has been previously utilized for colistin resistance, as reported in the literature. The authors are advised to provide a rationale for using CHCA, DHB, and super-DHB matrices in the discussion section.

References on usage of norhormane:

  1. https://doi.org/10.1128/spectrum.01445-21
  2. https://doi.org/10.1038/s41598-020-78401-3
  3. https://doi.org/10.1093/femspd/ftw097

Answer: Thank’s the Reviewer for the valuable comment. Indeed, norharmane is successfully using for bacterial lipids ionization and MALDI analysis, especially in the case of glycolipids such as lipid A. Nevertheless, the application of the norharmane matrix is accompanied by a negative ionization mode (as it was in suggested by the Reviewer works), while, in our study, we used the positive one. The use of the positive ionization mode was chosen since, besides the MALDI approach, we also used the SALDI technique and targets covered by silver nanoparticles where the positive ionization mode is preferable due to mechanisms of the ionization determined by silver nanoparticles. However, there is a possibility to use metal nanoparticles for ionization in negative mode since some NPs showed comparable efficiencies in negative ion mode (e.g., Ag for olefins, Ag and Au for sulfur compounds) [Lu M, Yang X, Yang Y, Qin P, Wu X, Cai Z. Nanomaterials as Assisted Matrix of Laser Desorption/Ionization Time-of-Flight Mass Spectrometry for the Analysis of Small Molecules. Nanomaterials (Basel). 2017 Apr 21;7(4):87. doi: 10.3390/nano7040087].  This approach provides interesting material for further studies on the application of the SALDI technique in the analysis of bacterial lipidomes. However, it is not the subject of the study of this paper. Regarding those matrices used in our studies, a suitable explanation has been added to Materials & Methods section.

Question 3: Comments on the Quality of English Language

Line 178:

Typo: The use instead of The used.

Answer: Done.

Reviewer 4 Report

Review on: „Silver nanoparticles targets fabricated using chemical vapor deposition method for differentiation of bacteria based on lipdomic profiles in laser desorption/ionization mass spectrometry“ by MaÅ›lak et al.

In their paper, Maślak and colleagues describe a lipodomic approach to bacterial species identification. The article is very well written, the experimental setup and the results comprehensible. A flow chart for the preparation of the samples and the performance of the analysis would nevertheless however be very helpful for the overall understanding (also gladly as a supplementary table). The authors should point out clearer why they chose E. coli in particular and chose to investigate cefuroxime. Furthermore, the authors should explain in more detail what is meant by the Folch method.

Author Response

Review on: „Silver nanoparticles targets fabricated using chemical vapor deposition method for differentiation of bacteria based on lipdomic profiles in laser desorption/ionization mass spectrometry“ by MaÅ›lak et al.

In their paper, MaÅ›lak and colleagues describe a lipodomic approach to bacterial species identification. The article is very well written, the experimental setup and the results comprehensible. A flow chart for the preparation of the samples and the performance of the analysis would nevertheless however be very helpful for the overall understanding (also gladly as a supplementary table). The authors should point out clearer why they chose E. coli in particular and chose to investigate cefuroxime. Furthermore, the authors should explain in more detail what is meant by the Folch method.

Answer: Thank's the Reviewer for the insightful reading of the paper. We provided a flow chart of the experiment's general workflow in the graphical abstract that included the most important steps and information to clearly understand the work's workflow.

Thank’s the Reviewer for the vital suggestion. A few reasons explain the choice of E. coli for this study. First,  E. coli is mentioned among the problematic bacterial species that are hard to differentiate from its close relatives - not only Shigella sp. but also E. coli subspecies differ in virulence level and antibiotic resistance, thus, posing a challenge for diagnostics laboratories and scientists. Secondly, E. coli is a model representative of Enterobacterales - the group demonstrated widespread resistance against beta-lactams, including cefotaxime (cephalosporins 3rd generation belonging to last-resort medicaments), which is, therefore, crucial for the health care system and from the epidemiological point of view. Since resistance against cefotaxime can be detected within Enterobacterales using the MALDI approach based on the indication hydrolysis of the antibiotic molecules, thus, the project aimed to check if the presence of such resistance can also be detected using other LDI techniques, which is an analysis of the bacterial lipidomic profiles.  Suitable explanation has been added to the manuscript – “E. coli is a model representative of Enterobacterales - the group demonstrated widespread resistance against beta-lactams, including cefotaxime (cephalosporins 3rd generation belonging to last-resort medicaments), which is, therefore, crucial for the health care system and from the epidemiological point of view. It is known that distinguishing  E. coli from its close relatives, including subspecies, differences in virulence level, and antibiotic resistance, can be problematic; however, as in the case of other Enterobacterales, resistance against cefotaxime can be detected using the MALDI approach based on the indication hydrolysis of the antibiotic molecules. Thus, the project aimed to check if such resistance can be detected using other LDI techniques, which analyze the bacterial lipidomic profiles.”

Regarding the Folch method used for lipids extraction, in the text, we briefly explain the principle of that method – “The extracts were prepared using the classical Folch method, a widely used standard method for lipid extraction in many laboratories. This method involves partitioning lipids in a biphasic mixture of chloroform and methanol”.